# Deep reaction network exploration at a heterogeneous catalytic interface

Qiyuan Zhao [1,2], Yinan Xu [1,2], Jeffrey Greeley [1] ✉ & Brett M. Savoie [1] ✉

Characterizing the reaction energies and barriers of reaction networks is central to catalyst development. However, heterogeneous catalytic surfaces pose several unique challenges to automatic reaction network characterization, including large sizes and open-ended reactant sets, that make ad hoc network construction the current state-of-the-art. Here, we show how automated network exploration algorithms can be adapted to the constraints of heterogeneous systems using ethylene oligomerization on silica-supported single-site $Ga^{3+}$ as a model system. Using only graph-based rules for exploring the network and elementary constraints based on activation energy and size for identifying network terminations, a comprehensive reaction network is generated and validated against standard methods. The algorithm (re)discovers the Ga-alkyl-centered Cossee-Arlman mechanism that is hypothesized to drive major product formation while also predicting several new pathways for producing alkanes and coke precursors. These results demonstrate that automated reaction exploration algorithms are rapidly maturing towards general purpose capability for exploratory catalytic applications.

Establishing the kinetic details of complex reaction networks is central to understanding heterogeneous catalytic surfaces[1-3]. The development of such networks for new systems is often painstaking, even when good hypotheses exist for the governing reactions and cycles[4,5]. Yet the domain knowledge necessary for hypothesis-driven exploration is often outpaced by the high throughput of synthetic and experimental efforts currently driving exploratory catalyst development. For these reasons, establishing reaction networks is typically a retrospective activity performed on already promising catalysts, rather than a predictive component of catalyst exploration workflows. To address this gap, it is urgent to develop computational methods to accelerate and automate the exploration, characterization, and refinement of complex reaction networks at surfaces[6].

In the context of heterogeneous catalysis, computational methods are relatively mature for characterizing the transition states (TSs) of targeted reactions[7-13], performing microkinetic modeling on established reaction networks, and using descriptor-based methods for optimizing catalysts[14-17]. However, a central challenge in characterizing new catalytic interfaces lies in establishing the kinetically relevant reaction network, which is often based on intuition and can be time-consuming and error prone to characterize ad hoc[4-6,18]. Even seemingly simple heterogeneous reactions, such as methane activation on metal oxide surfaces, can be decomposed into numerous elementary steps[19,20]. Catalytic cycles can also involve many intermediates, or even open-ended reactant lists, such that brute force enumeration and characterization are infeasible. Canonical examples of this include hydrocarbon forming reactions such as the oxidative coupling of methane and olefin oligomerization, each potentially involving the formation/dissociation of long carbon backbones as intermediates and an open set of olefins as adsorbed reactants[21,22].

For these reasons, the recent advent of automated reaction prediction approaches is potentially promising for elucidating reaction networks involving heterogeneous interfaces[23-25]. These methods can be categorized on the basis of whether the potential energy surface (PES) is explored in detail to locate TSs or whether the reaction networks are enumerated using a closed set of reaction templates. The latter class includes packages such as Network Generation (NetGen)[26] and Reaction Mechanism Generator (RMG)[27]. This strategy is less

[1]Davidson School of Chemical Engineering, Purdue University, West Lafayette, IN 47906, USA. [2]These authors contributed equally: Qiyuan Zhao, Yinan Xu. ✉e-mail: jgreeley@purdue.edu; bsavoie@purdue.edu

relevant to characterizing exploratory catalysts where established reaction templates are typically absent. Methods that directly explore the PES circumvent this limitation, at least in principle. This class includes several approaches that are under active development, including the artificial force induced reaction (AFIR) method[28,29], stochastic surface walking reaction sampling (SSW)[30], the ZStucture method[31,32], and Yet Another Reaction Program (YARP)[33], our recently developed methodology. All of these approaches are intrinsically more expensive than template-based methods because they sample the PES (e.g., using quantum chemistry calculations), which has been a major obstacle to applying them to heterogeneous systems in an exploratory context. For example, both SSW and AFIR have been applied to successfully (re)discover the relatively simple heterogeneous water-gas shift reaction occurring at a copper surface[29,34,35]. However, this analysis required millions of density functional theory (DFT) gradient calls, despite the small reactive system sizes. Due to the combinatorial scaling of possible reactions with the number of reacting atoms, a highly efficient reaction exploration scheme is indispensable to mitigate these costs. Heterogeneous applications also have several other technical obstacles to applying automated approaches, including the larger system sizes that are typical of surface models, the occurrence of spectator atoms that do not participate in reactions but nevertheless play important non-covalent or structural roles in the reaction pathways, and the use of periodic versus molecular models of the reacting systems. The optimal manner of overcoming these obstacles is an outstanding research question.

Here, we show how these problems can be addressed by combining a graph-based reaction exploration scheme, YARP, with a cluster model of a reactive interface. Ethylene oligomerization on silica-supported single site Ga³⁺ catalysts is used as a benchmark system for this approach based on the fact that, while considerable reaction data exists for this system, it still exhibits several unaccounted for product pathways. In particular, it has been previously reported that single site Ga³⁺ performs oligomerization chemistry

via the classic Ga-alkyl-centered Cossee-Arlman mechanism with reasonably high selectivity to short linear alpha-olefins at 250 ˚C and 1 atm. However, side products, such as light alkanes and coke, have also been detected, especially at higher temperatures, and the mechanistic details of these side pathways remain unexplored[36,37]. Here, YARP not only (re)discovers the established Ga-alkyl-centered Cossee-Arlman catalytic cycle producing 1-butene, but also discovers relatively low-barrier TSs for side reactions leading to the formation of isomers of 1-butene, odd-number oligomers, alkanes, and coke precursors. These pathways exhibit diverse mechanisms including carbon-backbone lengthening, oligomer liberation, and hydrogen transfer to form alkanes. The kinetic significance of the TSs is rationalized by detailed analysis of the energy surfaces of three representative catalytic cycles.

## Results and discussion

### YARP workflow

The YARP workflow for automatically exploring a reaction network on silica-supported single site Ga catalysts consists of four components: Ga-silica cluster model construction, graph-based product enumeration, transition state localization and characterization, and reaction cycle investigation (Fig. 1). In brief, a $Si_8O_{12}(OH)_8$ cluster is adapted from Ugliengo et al.[38]. A Ga-ethyl site is then created by substitution of a Si-OH moiety with a Ga atom and adding an ethyl group to the Ga site, which serves as the starting reactant (Fig. 1a). Once the cluster model is built, graph-based product enumeration is recursively applied to a subset of reactive atoms, involving the gallium, carbon, and hydrogen atoms attached to carbon in the cluster model (shown as pink, gray, and white spheres in Fig. 1b) to generate potential products. Product enumeration is performed using generic elementary reaction steps (ERS) consisting of bonding changes in the reactants that can be applied combinatorially to define closed reaction spaces. After the ERS-based product enumeration, transition states of the enumerated reactions are characterized using the same procedure as our earlier

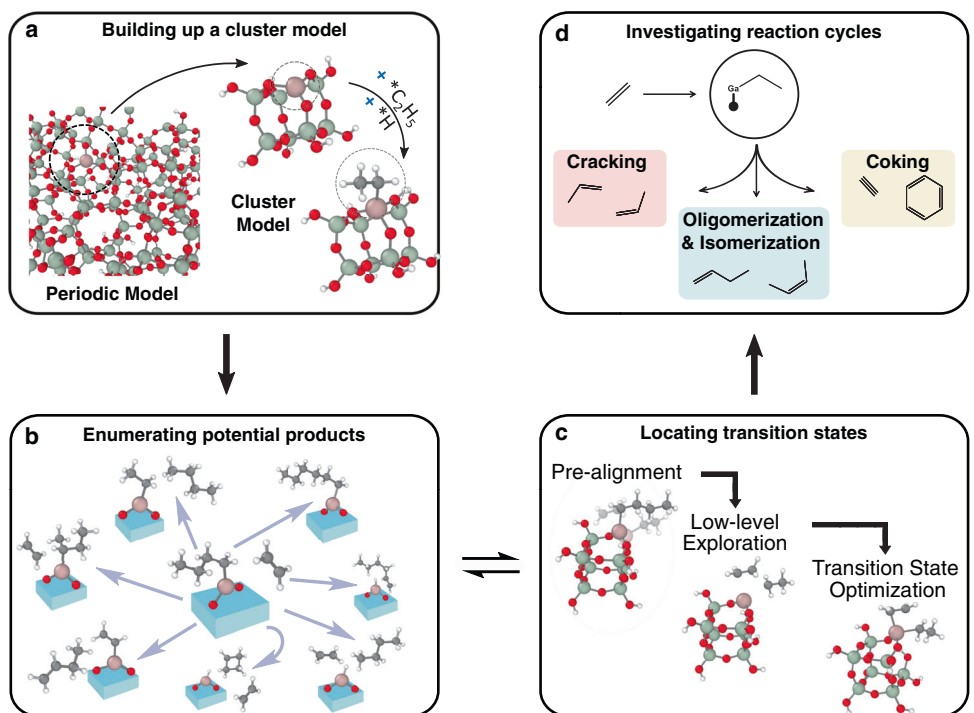

**Fig. 1 | Overview of automated reaction network characterization applied to ethylene oligomerization on single site Ga³⁺ catalysts supported on silica. a** A cluster model of a Ga³⁺ single site is built from a conventional periodic model. **b** Possible products are recursively enumerated from reactants/intermediates produced from elementary reaction steps on the cluster model. **c** Transition state localization and characterization are applied to each enumerated reaction. **d** Once the network exploration recursion terminates, detailed reaction mechanisms and relevant reaction cycles are summarized.

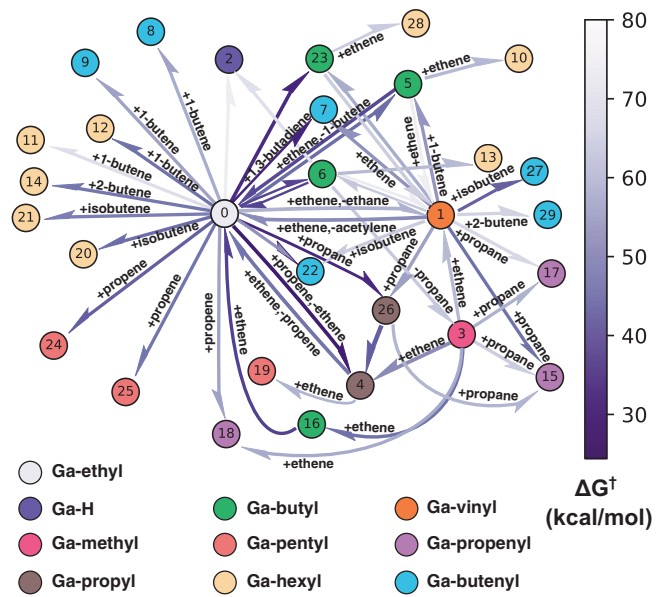

**Fig. 2 | Gallium catalyzed olefin oligomerization reaction network obtained from YARP exploration.** The edge colors reflect the activation free energy ($\Delta G^{\dagger}$) of each pathway as a measure of kinetic accessibility. Intermediate types are classified based on the alkyl and alkenyl attached to Ga and are denoted by different node colors.

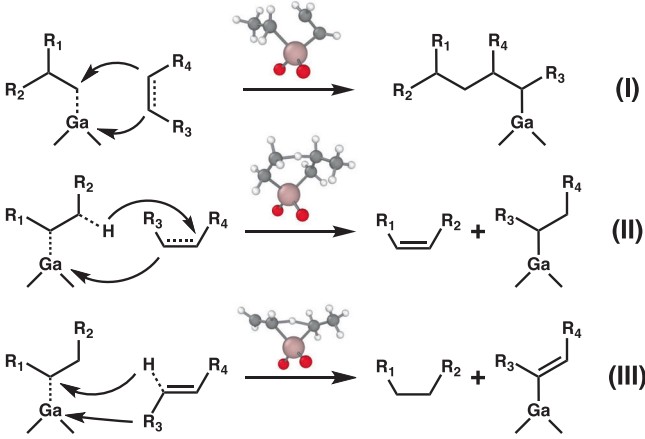

**Fig. 3 | Three elementary reaction types identified during reaction network exploration.** (I) olefin insertion; (II) $\beta$-hydride transfer; (III) $\alpha$-hydride transfer. $R_i$ refers to hydrogen, methyl, and ethyl groups. The presented TS structures represent the simplest examples of I-III reactions.

study[33], consisting of reaction geometry initialization and growing string method (GSM)[39] search performed with the GFN2-xTB semi-empirical model chemistry[40], followed by Berny optimization and intrinsic reaction coordinate (IRC) calculations performed using DFT (Fig. 1c). All of the kinetically-relevant products explored in this procedure serve as inputs for further reaction exploration until the discovery of new reactions is exhausted. In the end, important reaction pathways and catalytic cycles are analyzed in detail based on the reaction mechanisms discovered during the exploration (Fig. 1d). Detailed descriptions of each component are provided in the Methods section.

## The complete reaction network involving Ga³⁺

The overall reaction network that was generated by YARP for ethylene oligomerization on silica-supported single site Ga³⁺ is shown in Fig. 2. Network exploration was initialized with the Ga-ethyl species (node 0 in Fig. 2), which has been proposed as a key intermediate in the Cossee-Arlman ethylene oligomerization cycle[37]. After a single step of reaction enumeration and TS characterization, Ga-n-butyl, Ga-vinyl + ethane, and Ga-hydride + 1-butene, were identified as intended products of reactions between Ga-ethyl and ethylene. The free energies of activation ($\Delta G^{\dagger}$) of forming Ga-n-butyl, Ga-vinyl, and Ga-hydride are 44.1, 59.8, and 93.5 kcal/mol, respectively. Based on its high activation energy of formation, YARP excluded Ga-hydride from further exploration, whereas Ga-n-butyl and Ga-vinyl were included as active nodes for further reaction exploration. The high activation energy of $\beta$-hydrogen elimination to form Ga-hydride has also been observed in previous studies using conventional periodic DFT analysis[36,37]. The second step of exploration identifies Ga-n-butenyl (from Ga-vinyl, $\Delta G^{\dagger} = 53.2$ kcal/mol), acetylene (formed with Ga-ethyl from Ga-vinyl, $\Delta G^{\dagger} = 51.4$ kcal/mol), Ga-hexyl (from Ga-butyl, $\Delta G^{\dagger} = 61.3$ kcal/mol), 1-butene (formed with Ga-ethyl from Ga-butyl, $\Delta G^{\dagger} = 36.0$ kcal/mol) and butane (formed with Ga-vinyl from Ga-butyl, $\Delta G^{\dagger} = 76.4$ kcal/mol) as intended products. Notably, the lowest barrier step yielding 1-butene constitutes a rediscovery by the algorithm of the classic Cossee-Arlman mechanism that has previously been studied as the likely pathway for major product formation in this system. Based on

the activation energies of the reactions at this iteration, Ga-n-butenyl was included as a new active node for further exploration (node 7), Ga-n-hexyl was classified as a terminal node (node 13) due to its size (see Methods for termination criteria), and 1-butene was added to the free-olefin list as a candidate for further reactions with the active nodes, Ga-ethyl (node 0) and Ga-vinyl (node 1). YARP recursively explored the reaction space via the same approach that was employed in the first and second iteration until all reactions within the prescribed constraints had been explored. All reactions explored with $\Delta G^{\dagger} < 80$ kcal/mol are presented in Fig. 2, and detailed geometries of each node can be found in the SI.

## Three key reaction types occurring on Ga³⁺

Three distinct types of reactions were discovered during the network exploration that are distinguished by their reactions with the adsorbed carbon species. All instances of each class exhibit $\Delta G^{\dagger} < 70$ kcal/mol. The first type is responsible for lengthening (or breaking, in the case of the reverse reaction) the carbon backbone (Type I in Fig. 3). The TS of the Type I reaction involves a "C=C" moiety bonding to the catalyst to form a four-coordinated Ga intermediate that precedes bond formation with an adsorbed alkyl species. The second type of reaction is $\beta$-hydride transfer that liberates an oligomer and closes an oligomerization cycle (Type II in Fig. 3). In the TS of this reaction type, the $\beta$-hydrogen of the adsorbed alkyl species transfers to an incoming olefin, which binds to the Ga center and becomes a new adsorbate. An oligomerization cycle can also be completed by a $\beta$-hydride elimination step to form Ga-hydride, but YARP predicts a much higher activation energy for this pathway (see Supporting Information for details). The facile $\beta$-hydride transfer step on Ga is a fundamentally different reaction channel from those occurring on traditional transition metal catalytic sites, where the hydrogen being transferred is not interacting with the metal center[41]. The third type of reaction produces an alkane, leaving a hydrogen-deficient adsorbed species, like Ga-vinyl (Type III in Fig. 3). The TS of the Type III reaction resembles that of Type II, except that the hydrogen transfers to the $\alpha$-carbon. Alkane formation has been reported in multiple olefin oligomerization experiments[36,41–43], which may be explained by moderate barrier Type III pathways. Further, we hypothesize that the products of type III reactions may undergo additional type I and type II steps. The combination of type I-III reactions may eventually liberate alkynes and aromatics that are commonly considered to be coke precursors[44,45]

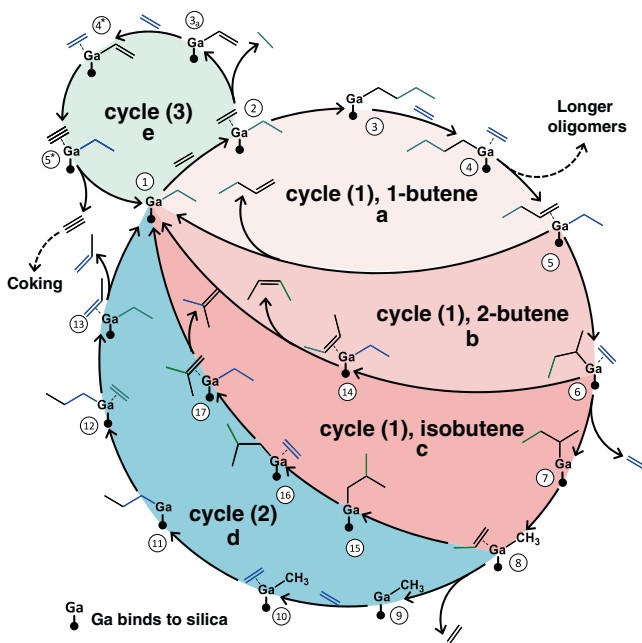

**Fig. 4 | Proposed pathways of oligomerization (light red), isomerization (medium and dark red), cracking (blue), and coking (green) reactions. a** Oligomerization pathway to 1-butene. Isomerization pathways forming (**b**) 2-butene and **c** isobutene. **d** Cracking pathway for propylene formation. **e** An example of a pathway forming alkanes and hydrogen-deficient products starting from Ga-ethyl. Similar alkane formation cycles can also occur for species ②, ④, ⑥, and ⑩.

(region e in Fig. 4). For example, a low barrier (~42 kcal/mol) Ga-catalyzed conversion of ethyne to benzene is compared with a non-catalyzed conversion in Fig. S3.

In addition to participating in the expected Cossee-Arlman oligomerization cycle, the elementary reaction types described above can contribute to several catalytic cycles for olefin isomerization and chain cracking (Fig. 4). In particular, following the formation of 1-butene and the recovery of the Ga-ethyl intermediate (species ⑤ in Fig. 4), where a Cossee-Arlman oligomerization cycle is nearly complete, the 1-butene molecule can be re-adsorbed with a simple rotation to react with Ga-ethyl through another $\beta$-hydride transfer (type II), producing Ga-2-butyl (species ⑥). This newly reported intermediate can undergo a facile type II reaction, forming cis- or trans-2-butene (only cis-2-butene formation is considered here, species ⑭). Butenes have been previously detected in experiments at 523 K and 1 atm, but distinguishing between cis and trans isomers is challenging due to their similar retention times in gas chromatography[36]. In an alternative pathway, Ga-2-butyl can undergo additional type I and II reactions to form Ga-methyl with physisorbed propylene (species ⑧). In addition, there can be another re-adsorption step of propylene on Ga-methyl, resulting in a Ga-isobutyl species (species ⑮), which eventually leads to isobutene (species ⑰). Throughout the isomerization and cracking pathways, the type III step can occur on each Ga-alkyl species, including Ga-ethyl, Ga-propyl, and Ga-butyl species. For example, a plausible pathway involving the type III reaction is outlined in the green circle of Fig. 4, where the resulting Ga-vinyl intermediate undergoes additional $\beta$-hydride transfer, leading to the formation of acetylene (a coke precursor).

**Kinetic significance of type I-III transition states**

Potential energy diagrams were used to compare the kinetic relevance of reactions cycles discovered for oligomerization, isomerization,

cracking, and coking pathways (Fig. 5):

$$2C_2H_4 \longrightarrow C_4H_8 \tag{1}$$

$$3C_2H_4 \longrightarrow 2C_3H_6 \tag{2}$$

$$\frac{n+2}{2}C_2H_4 \longrightarrow C_2H_2 + C_nH_{2n+2}, \text{ where } n=1, 2, \text{ and } 3. \tag{3}$$

The competition between these cycles determines the selectivity of producing gaseous products and coke precursors. Cycle (1) involves ethylene dimerization products (Fig. 4a–c), including 1-butene and associated isomers. One catalytic cycle closes through an ethylene insertion (denoted as type I) and a $\beta$-hydride transfer (denoted as type II). Following additional type I-II steps occurring on Ga-ethyl with an adsorbed 1-butene (species ⑤), cis-2-butene and isobutene can also form (Fig. 4b, c). Cycle (2) involves the formation of cracking products, such as propylene, which are not favorable at a relatively low temperature (250 ℃, 1 atm). One propylene molecule can be obtained through C-C bond breaking of a Ga-2-butyl species (reverse type I). The production of a second propylene molecule occurs via the same Cossee-Arlman oligomerization cycle initiated by the Ga-methyl intermediate (species ⑨, Fig. 4d). In cycle (3), the type III elementary step generates an alkane, which may occur for all Ga-alkyl intermediates, and an alkyne, such as acetylene, is formed that balances the stoichiometry. A relatively facile acetylene formation pathway occurs through a type II step occurring on the Ga-vinyl species from the type III reaction (species ⓪, Fig. 4e). Many other relatively low barrier pathways (≤70 kcal/mol) are discovered by YARP, including the formation of various $C_nH_{2n}$ species, and $C_nH_{2n-2}$ isomers. As discussed later, although these pathways are not as favorable as the primary Cossee-Arlman cycle, they may be responsible for coking and deactivation pathways and result in a broad diversity of possible products.

To validate the accuracy of the cluster model results, they were benchmarked against conventional periodic DFT with the NEB-Lanczos algorithm for localizing TSs of reaction cycles (1) and (2) (Fig. 5a). For this comparison, the geometries and energies for the cluster model were recalculated at the B3LYP-D3/6-311G(d,p) level of theory to minimize the DFT errors as a confounding factor when comparing the cluster and slab results (see the Supporting Information for additional details on selecting a suitable DFT level). Overall, periodic DFT and the cluster model predict similar binding energies, reaction energies, and reaction barriers, while some systematic deviations can be observed, including modestly higher activation energies predicted by the cluster model. For example, the cluster and periodic models predict activation energies of 1.8 and 1.6 eV, respectively, for the ethylene insertion step (species ② to species ③). The difference may be attributed to long-range order and reconstruction effects in the silica support, which may systematically lower activation energies, but are absent in the cluster model. Another systematic difference is that the binding energies obtained from the periodic model are consistently lower (less negative) than those obtained from the cluster model (Fig. 5a). In the periodic model, the Ga site is surrounded by siloxane frameworks with various ring sizes, which can contribute to a weakened binding due to steric effects, especially for the species with longer carbon chains or with physisorbed olefins. This effect is particularly significant for species ⑤, where 1-butene, the largest physisorbed reactant in our analysis, is involved. The comparison is also affected by the distinct functionals that were used due to their differing availability in the reference molecular and periodic quantum chemistry packages. Nevertheless, the two approaches predict similar relative barriers for all of the TSs under each elementary step type. The mean difference between activation energies for type I versus type II reactions are both

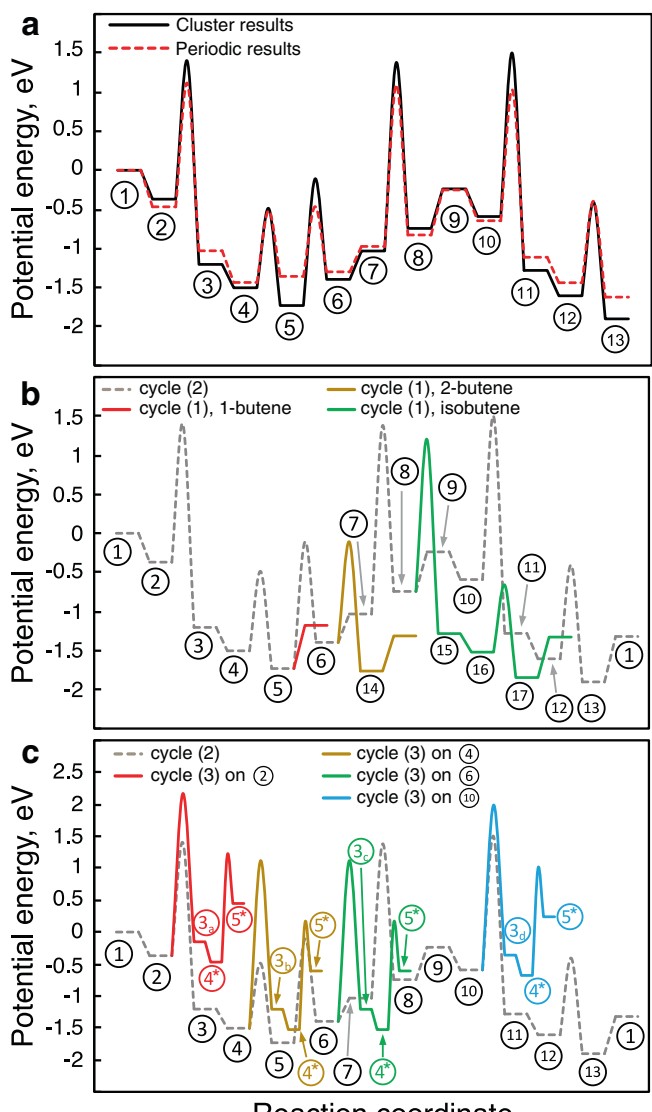

**Fig. 5 | Energy diagrams of three kinetically relevant reaction cycles discovered within the reaction network. a** Comparison of the energy landscape for cycle (2) using the cluster model and periodic slab. **b** Comparison of competing olefin formation pathways (colored) and cracking pathways (gray). **c** Comparison of competing acetylene formation pathways (colored) and cracking pathways (gray). The species are numbered based on the pathway diagram in Fig. 4, and the energies are reported with respect to the single point energy of Ga-ethyl plus a gaseous ethylene molecule.

0.8 eV, calculated by the cluster model and the periodic model. Further, all type I transition states are similarly accessible (i.e. barriers within 0.1 eV), and both models predict type II reactions to have consistently lower barriers. The overall agreement between the cluster and periodic models with respect to relative barrier heights validates the usefulness of the cluster models for performing reaction network exploration on silica surfaces.

Figure 5b, c outline the energy landscape comparison between the overall reaction cycles (1)-(3) using the cluster results. In cycle (1), where the carbon chain length doubles and 1-butene is formed (species ① - ⑤), the ethylene insertion involves a higher activation energy (1.76 eV) than the olefin liberation step (1.02 eV). This is consistent with a previous study showing that ethylene insertion is rate-determining in this system[37]. In the energy landscape of cycle (2), three type I elementary steps have relatively high activation energies: the ethylene insertion shared with cycle (1), the cracking of Ga-2-butyl (species ⑦,

2.40 eV), and the step forming Ga-1-propyl (species ⑩) from Ga-methyl and ethylene (2.08 eV). The cracking of Ga-2-butyl to form Ga-methyl and propylene involves the highest activation energy since it is a reversed type I step. Both periodic and YARP-cluster results predict that type I reactions are exothermic. Therefore, cycle (2) will not dominate the reaction network. Indeed, previous experimental results of Ga single sites show a strong selectivity to olefin oligomerization at 250 °C and 1 atm, forming short linear oligomers[36]. However, the activation free energy of cracking reduces as temperature increases (due to the entropically favored reverse type I step) in comparison to the formation of longer Ga-alkyl carbon chains, thus narrowing the energy difference between Ga-2-butyl (species ⑦) and the cracking TS. Based on the reaction entropies, the selectivity for propylene production over butene should increase with increased temperature or reduced pressure since cycle (2) produces a higher number of gas molecules than cycle (1). Finally, the high barrier of the reverse type I step provides a basis for the competition between type I and III reactions starting from the Ga-2-butyl species. In particular, the formation of 1-butane (species ⑥ - ⑩) can be competitive with cracking reactions (species ⑦ - ⑧). Subsequently, acetylene formation can occur via facile type II reactions (species ⑪ - ⑩, 1.68 eV). Therefore, our pathway analysis suggests that type III reactions are kinetically less favorable, but nevertheless represent side-reaction channels that become accessible as they compete with the reverse of type I step. With the formation of alkynes, other side reactions, such as aromatization and coking, may occur as subsequent thermodynamic products, especially at higher temperatures[46,47]. For example, the selectivities for producing alkynes, aromatics, and coke may increase with the pressure of ethylene since it shifts the reaction free energy of acetylene condensation to benzene. These insights into the reaction network and TSs downstream of Ga-ethyl formation also generalize to other metal single sites with similar electronic properties, such as Zn and Al[48,49]. In the Supporting information section 2, we have included the analogous energy diagrams for the reactions involving an Al single site. These results show energy barriers and a competition between cycles (1)-(3) that are similar to those of Ga.

Given the generic reaction rules and size constraints that were used to generate the ethylene oligomerization reaction network in this work, there are many opportunities for applying this approach to other heterogeneous systems. Among the salient details of the implementation to consider for future applications are the use of a cluster model as a surrogate for a periodic slab and the major speedup provided by semi-empirical quantum chemistry. Neither detail is intrinsic to applying YARP, and indeed, the cluster assumption was validated here and adopted out of convenience. There are, however, no obstacles to applying YARP using a periodic code, outside of cost. The applicability of this approach to other heterogeneous surfaces is therefore anticipated and is currently under investigation.

These results demonstrate how automatic exploration can be applied to heterogeneous catalytic networks using ethylene oligomerization catalyzed by a silica-supported Ga single site as a benchmark. The method (re)discovered the classic Ga-ethyl-centered Cossee-Arlman oligomerization cycle and several side-product pathways using generic graphical reaction rules and an ultra-low cost TS localization framework. The reaction network elaborated here represents the largest (in terms of both the number of reactive atoms, 20, and the branched alkyl intermediates up to $C_6$) that has been reported for a heterogeneous catalyst using an automated exploration algorithm based on quantum chemically characterized transition states. We do not expect this record to last, since there are relatively few impediments to applying this approach to other systems. Among the foreseeable challenges are that we have not considered adsorption and desorption steps that are often rate-limiting within catalytic cycles. When these steps are non-covalent and molecular in nature, the presented framework can already support them in scenarios where they

apply. A more fundamental limitation is that the catalytic surface was treated as static, save for the reactive atom and its nearest neighbors. This assumption will break down for cycles where the surface significantly restructures, such as for nanocatalysts, or where electrodeposition/desorption occurs. This could be accommodated by expanding the space of reactive atoms, but this will increase costs and presents opportunities for further innovation.

As network exploration becomes a predictive tool for catalyst screening, analyzing the reaction data and extracting general insights will become a bottleneck. As one example, characterizing the full reaction network reported here took around one-week of computational time on minimal resources (one node with 128 cores), but the costly manual process of classifying reaction mechanisms and closed cycles belonging to the same family of transformations took the majority of the time in this study. Although knowledge extraction of this kind will always be manual to a degree, there are several opportunities for streamlining this through automated mechanism classification based on orbital analyses and microkinetic modeling, both of which will be leveraged in the future.

## Methods

### Ga-silica cluster model construction
Ethylene oligomerization on single-site $Ga^{3+}/SiO_2$ was modeled based on a $Si_8O_{12}(OH)_8$ cluster that was adapted from Ugliengo et al.[38], wherein a $Ga^{3+}$ single-site was created by substitution of a Si-OH moiety with a Ga atom. A Ga-ethyl site was, in turn, created by adding an ethyl group to the Ga site and a proton to the adjacent oxygen atom to maintain charge balance (Fig. 1a). The focus of this effort was to establish the reaction network downstream of Ga-ethyl formation since this is where several mechanistic gaps exist in the catalytic cycle, especially with respect to side-product formation. In particular, previous work has established facile initiation of the Ga-ethyl species from a bare single site and gaseous ethylene, as well as a low-barrier Ga-ethyl-centered Cossee-Arlman pathway to ethylene oligomerization. We have previously examined the competing Ga-hydride-centered Cossee-Arlman cycle and the empty Ga site-centered proton transfer cycle, and facile formation of the Ga-ethyl was observed in all cases[37]. Nevertheless, the choice to use Ga-ethyl as a starting point for the exploration is non-essential, since the network exploration used here automatically discovered all possible $C_1$-$C_4$ Ga sites and gas-phase olefins during the reaction network exploration.

The cluster model can be viewed as a finite portion of the solid silica surface, with the dangling oxygen atoms passivated by hydrogen atoms[38,50]. The localized nature of oxides and the $Ga^{3+}$ center make the cluster model a credible approximation for assessing surface reactivity, as demonstrated by our previous analysis on ethylene oligomerization[37]. Cluster models provide a potentially useful bridge between the active methods development occurring for molecular reaction network exploration and reaction networks occurring at reactive surfaces. Here, comparisons of the energies and barrier heights calculated on periodic surfaces and the cluster model were used to further validate this assumption (typical differences in activation energies were around 0.3 eV, Table S2).

### Reaction network characterization
The YARP methodology was used to enumerate the reactions and characterize the TSs associated with the Ga-ethyl species modeled in the presence of excess ethylene. For a more detailed description of the YARP methodology, we direct readers to our previous publication[33]. In the following sections, we focus on the modifications that were implemented to the reaction enumeration and reaction pathway construction steps to adapt YARP to explore ethylene oligomerization on single-site $Ga^{3+}/SiO_2$. The salient features of YARP are that all exploration steps are automated and based on quantum chemically characterized transition states rather than particular user-defined reaction mechanisms. For these reasons, minimal human intervention is required for the network exploration after the initial setup of the reacting system and the termination criteria.

**Product enumeration.** The YARP methodology consists of recursively applying graph-based elementary reaction steps (ERS) of the form break *m* bonds and form *n* bonds (bmfn). These rules are sufficiently generic to recapitulate many reactions without relying on explicit reaction templates, and they define reaction spaces that can be comprehensively explored (as an example, all b2f2 pathways of a given set of reactants constitute a well-defined set). For neutral closed-shell systems, the simplest reaction that yields non-trivial closed-shell products is b2f2 (describing, for example, an E2 reaction); however, single-step b3f3 reactions, such as Diels-Alder and Claisen rearrangements, are also common. Here, we applied a compromise scheme, including all b2f2 reactions and the subset of b3f3 reactions that involved at least one double-bond breaking. These ERSs were applied to the gallium, carbon, and hydrogen atoms attached to carbon in the cluster model (shown as pink, gray, and white spheres in Fig. 1b) to enumerate all products for each reactant in the network. Reactions that did not involve Ga (e.g., non-catalytic reactions between ethane and other alkyl products), and reactions that yielded species with more than five carbons were discarded from consideration.

**Transition state localization.** After product enumeration, YARP attempts to localize TSs for each reaction. This procedure consists of initializing a reaction geometry, estimating the TS geometry at the semi-empirical GFN2-xTB[40] level using the growing string method (GSM)[51], TS optimization at the DFT level using Berny optimization, and intrinsic reaction coordinate (IRC) calculations to classify the resulting TSs (Fig. 1c). For the geometry initialization, the joint-optimization algorithm reported in the original YARP publication was retained, with the exception that the positions of all silica atoms except the two oxygen atoms attached to gallium were fixed to preserve the initial DFT-level cluster structure. The joint-optimization algorithm is designed to produce minimally displaced reactant-product conformers that are well-conditioned for a double-ended TS search[33]. These optimized structures were then used as the fixed endpoints for GSM calculations. After convergence, the highest energy node along the reaction pathway was selected as the initial guess for an unconstrained DFT level Berny transition state optimization. The final TS, after successful convergence of previous steps with a structure exhibiting a single imaginary frequency, was characterized by an IRC calculation to ensure its correspondence to the attempted reaction. When the two end nodes obtained by the IRC calculation matched the input reactant and product, the attempted reaction was classified as "intended" and included in the reaction network.

**Reaction network construction.** To construct the reaction network, interleaved product enumeration and TS localization was performed until the discovery of new reactions was exhausted. At each stage of this iteration, the Ga-products of the previous iteration served as potential reactants for the next iteration subject to conditions that were designed to manage the size of the reaction network while being relatively permissive in terms of exploring new reactivities. Specifically, Ga-adsorbed species were only included as potential reactants at the next iteration if they were connected to the rest of the network by an intended reaction with an activation energy less than 3 eV (~70 kcal/mol, above which the reaction will require extremely high temperatures to be kinetically competitive, and under which conditions thermal reactions will, in any case, be dominant[52,53]). It is possible for a Ga-adsorbed species to fail the activation energy constraint at an early iteration, but then to be included later if an alternative pathway is discovered. Additionally, the size of the reactant species attached to the gallium site was limited to butyl and smaller moieties to avoid the

trivial growth of the network due to lengthening of the carbon backbone. All of the Ga intermediates obtained without violating these constraints were included as species capable of participating in reactions in the next iteration. At each iteration, the set of explored reactants consisted of all combinations of the active Ga-adsorbed species and any free olefins that were produced as products during previous iterations of exploration. Thus, a newly generated Ga-adsorbed species would participate in up to $n+1$ separate reactant sets, where $n$ is the collection of free olefins discovered up until that point of exploration, and the plus one corresponds to unimolecular reactions involving the Ga-species. Reactant combinations involving more than six carbons were discarded to avoid uninteresting growth of the network. For each set of reactants, the ERS-generated reactions were characterized, and the recursion ended after no new reactions were discovered.

### Periodic DFT calculations

The reaction energies and reaction barriers for a subset of pathways were recalculated on an amorphous silica slab model with a large unit cell ($21.6\ Å \times 21.6\ Å \times 34.5\ Å$) and compared with cluster model results for validation. These calculations were performed on the amorphous structure reported by Comas-Vives, generated from an annealing process using classical molecular dynamics and multiple dehydration processes that result in a high level of dehydroxylation (1.1 silanol moieties $nm^{-1}$) and siloxane rings with different sizes[54]. The Ga-ethyl moiety was created using the same approach as was employed in the cluster models. Previous studies indicate that the less-constrained, three-coordinated Ga sites are responsible for the oligomerization chemistry, whereas the constrained four-coordinated sites are relatively inactive due to strong steric hindrance effects[37]. Therefore, the periodic slab calculations focused on the less constrained Ga site, as represented in our cluster model.

### Computational details

YARP used Gaussian 16 as the reference quantum chemistry engine for the DFT calculations associated with the Berny optimizations and IRC calculations[55]. Calculations were performed at the B3LYP/6-31G level of theory during network exploration. The geometries of the TSs, reaction energies, and reaction barriers were recalculated at the B3LYP-D3/6-311G(d,p) level of theory for validation and comparison with periodic calculations. This level of theory showed a balance of accuracy and cost describing the $Ga^{3+}/SiO_2$ system (Figs. S1 and S2). The GSM calculations were performed by the pyGSM package using eleven images, fixed reactant and product geometries, and other default hyperparameters[39]. All GFN2-xTB calculations were performed with the xTB program (version 6.2.3)[40].

Periodic DFT calculations were performed using Vienna Ab-initio Simulation Package (VASP, 5.4.1), where planewave basis sets describe the Kohn-Sham orbitals, and the Kohn-Sham equations were solved self-consistently[56–60]. The BEEF-VdW exchange-correlation functional with projector augmented wave (PAW) pseudopotentials was employed[60–62]. A Monkhorst-Pack $k$-sampling was used, and a $k$ point grid of $2 \times 2 \times 1$ was applied. A cutoff energy of 400 eV and a force-convergence criterion of 20 meV $Å^{-1}$ for energy local minima were used. The climbing image nudged elastic band (CI-NEB) method was used as a first step to locate transition states[63,64]. Seven images were used in each NEB calculation, as generated by the Image Dependent Pair Potential (IDPP) tool[65]. Following each NEB calculation, Lanczos diagonalization was used to identify the transition state with a greater accuracy[66]. The force convergence criterion of 20 meV $Å^{-1}$ was used for TS optimization. All TSs were confirmed to exhibit only one imaginary frequency.

## Data availability

The authors declare that the data supporting the findings of this study are available within the paper and its supplementary information files. Source data for Figs. 2 and 5 are available in Source Data. The raw data generated in this study have been deposited in the YARP database and are available at https://doi.org/10.6084/m9.figshare.14766624[67], including raw output files and molecular (reactants, products and transition states) geometries.

## Code availability

The version of YARP used in this study is available through GitHub under the GNU GPL-3.0 License [https://github.com/zhaoqy1996/YARP]. The specific version of the package used to generate the results in this study can be found at https://doi.org/10.5281/zenodo.6828628 [68].

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

## Acknowledgements

The work performed by Q.Z. and B.M.S was made possible by the Office of Naval Research (ONR) through support provided by the Energetic Materials Program (MURI grant number: N00014-21-1-2476, Program Manager: Dr. Chad Stoltz). Y.X. and J.G. acknowledge support from the National Science Foundation through the Center for Innovative and Sustained Transformation of Alkane Resources (CISTAR) under Cooperative Agreement No. EEC-1647722. Use of the Center for Nanoscale Materials, an Office of Science user facility, was supported by the U.S. Department of Energy, Office of Science, Office of Basic Energy Sciences, under Contract No. DE-AC02- 06CH11357. Use of the National Energy Research Scientific Computing Center is also gratefully acknowledged. Q.Z., Y.X., J.G., and B.M.S. would like to thank Prof. Jeffrey T. Miller from Purdue University and Dr. Nicole J. LiBretto from National Renewable Energy Laboratory (NREL) for the insightful discussions.

## Author contributions

The project was conceived by Q.Z., Y.X., J.G., and B.M.S. Q.Z. and Y.X. designed and performed the simulations. Y.X. established the structures of catalytic sites on cluster and periodic silica models. Q.Z. performed the automated reaction network exploration and the related cluster DFT calculations. Y.X. performed the periodic DFT calculations. All authors analyzed the results and prepared the manuscript.

## Competing interests

The authors declare no competing interests.
