## [Peer Review File · Nature Communications]

Title: Deep Reaction Network Exploration at a Heterogeneous Catalytic InterfaceEditorial Note: This manuscript has been previously reviewed at another journal that is not operating a transparent peer review scheme. This document only contains reviewer comments and rebuttal letters for versions considered at *Nature Communications*.

REVIEWERS' COMMENTS

Reviewer #1 (Remarks to the Author):

The authors have done a good job in addressing my previous comments - I feel that the manuscript is overall clearer in defining its main contribution.

However, I do still feel that my previous comment #1 still stands somewhat - in particular, I noted that I believe that the unique aspect regarding the size of the system studied here is not particularly strong. Following my previous review, the authors have added the clause that their network is based on transition-state calculations - I agree that this is an important distinction to be made. For the other studies previously referenced that did perform TS optimization for heterogeneous systems, the authors note that these systems were previously small - and it seems that their simulations are the largest (up to 20 atoms) available to date.

While I agree that this does appear to be the case (at least as far as I know of the literature in this field), it feels like we're now down to splitting hairs about the topic. It's not difficult to find previous reaction discovery simulation studies that are much larger than the systems considered here and identify the transition-states in the generated networks - for example, I believe that the ab initio nanoreactor work of Todd Martinez and coworkers (e.g. 10.1021/acs.jctc.5b00830, plus the original nature chem. report - also ACS Cent. Sci. 2019, 5, 1532–1540) considers much larger reactive systems than that studied here. Zimmerman has also adapted his Zstruct method to consider surface reactions (Phys.Chem.Chem.Phys., 2018, 20, 7721), and Maeda has also adapted AFIR to study surfaces (Phys.Chem.Chem.Phys., 2019, 21, 14366).

So, the bottom line is that the system size considered is not a unique aspect when compared to previous reaction-discovery simulations - and previous reaction discovery simulations have been performed for heterogeneous systems too, including using methods that employ transition-state searches. So, it seems that the main remaining aspect of novelty in the simulations is that the system considered here (up to 20 active atoms) is larger than previous studies of heterogeneous systems employing TS searches (again - it is not particularly large compared to many other studies performed by methods such as ab initio nanoreactor). My remaining comment is that the authors should therefore be clear about these distinctions about system sizes previously treated by different methodologies.

Reviewer #2 (Remarks to the Author):

The authors addressed my comments and questions satisfactorily. I think the paper improved compared to the former version and is suitable for publication in Nature Communications.

A point-by-point response to the reviewer comments is below. We have attached a highlighted version of the revised manuscript showing material that has been added or removed with yellow highlights and strikethroughs, respectively. To keep the length of the response letter reasonable, we refer to the line numbers in the highlighted version of the manuscript to indicate the changes.

Reviewer #1 (Remarks to the Author):

The authors have done a good job in addressing my previous comments - I feel that the manuscript is overall clearer in defining its main contribution.

Response: We appreciate the time taken for a second review and the overall supportive reading of the work.

However, I do still feel that my previous comment #1 still stands somewhat - in particular, I noted that I believe that the unique aspect regarding the size of the system studied here is not particularly strong. Following my previous review, the authors have added the clause that their network is based on transition-state calculations - I agree that this is an important distinction to be made. For the other studies previously referenced that did perform TS optimization for heterogeneous systems, the authors note that these systems were previously small - and it seems that their simulations are the largest (up to 20 atoms) available to date.

While I agree that this does appear to be the case (at least as far as I know of the literature in this field), it feels like we're now down to splitting hairs about the topic. It's not difficult to find previous reaction discovery simulation studies that are much larger than the systems considered here and identify the transition-states in the generated networks - for example, I believe that the ab initio nanoreactor work of Todd Martinez and coworkers (e.g. 10.1021/acs.jctc.5b00830, plus the original nature chem. report - also ACS Cent. Sci. 2019, 5, 1532–1540) considers much larger reactive systems than that studied here. Zimmerman has also adapted his Zstruct method to consider surface reactions (Phys.Chem.Chem.Phys., 2018, 20, 7721), and Maeda has also adapted AFIR to study surfaces (Phys.Chem.Chem.Phys., 2019, 21, 14366).

So, the bottom line is that the system size considered is not a unique aspect when compared to previous reaction-discovery simulations - and previous reaction discovery simulations have been performed for heterogeneous systems too, including using methods that employ transition-state searches. So, it seems that the main remaining aspect of novelty in the simulations is that the system considered here (up to 20 active atoms) is larger than previous studies of heterogeneous systems employing TS searches (again - it is not particularly large compared to many other studies performed by methods such as ab initio nanoreactor). My remaining comment is that the authors should therefore be clear about these distinctions about system sizes previously treated by different methodologies.

Response: We don't want to give the impression that we are splitting hairs (to use the reviewer's phrase), while at the same time we think that it is fair to maintain the distinctions of the current approach with respect to size and chemical scope. With respect to the three examples given, we discussed them all after the original submission. In addition to the reasons given previously, the nanoreactor is inappropriate for surfaces since the caustic reaction conditions would obliterate any

surface that it was used to study. We feel like it would be impolitic to go out of our way to say this in the main text and so have not included it in the introduction since it isn't a relevant comparison (even if it is well known). The other two methods mentioned by the reviewer (AFIR and ZStruct) are both discussed in the main text and are applied to much smaller systems in both of the references provided.

As a middle ground, we have removed the claim about "largest" from the abstract, since that is the crux of the reviewer's concern and this claim is ultimately inessential to the impact of the work. We have also added a clarification about the need for improvements beyond AFIR and ZStruct and two of the references provided by the reviewer to the introduction.

Abstract: We have removed the original claim regarding system size from the abstract and replaced it with a more general statement: "These results demonstrate that automated reaction exploration algorithms are rapidly maturing towards general purpose capability for exploratory catalytic applications."

Line 31: We have added the ZStruct and AFIR references provided by the reviewer.

Line 39: We have clarified the combinatorial challenge of dealing with large reactive spaces.

Reviewer #2 (Remarks to the Author):

The authors addressed my comments and questions satisfactorily. I think the paper improved compared to the former version and is suitable for publication in Nature Communications.

Response: We appreciate the time taken for a second review and the positive assessment of the work.